# Ultra-fast vortex motion in a direct-write Nb-C superconductor

O. V. Dobrovolskiy 1,2✉, D. Yu Vodolazov 3,4, F. Porrati 5, R. Sachser5, V. M. Bevz2, M. Yu Mikhailov 6, A. V. Chumak 1 & M. Huth 5

The ultra-fast dynamics of superconducting vortices harbors rich physics generic to none-quilibrium collective systems. The phenomenon of flux-flow instability (FFI), however, prevents its exploration and sets practical limits for the use of vortices in various applications. To suppress the FFI, a superconductor should exhibit a rarely achieved combination of properties: weak volume pinning, close-to-depairing critical current, and fast heat removal from heated electrons. Here, we demonstrate experimentally ultra-fast vortex motion at velocities of 10–15 km s$^{-1}$ in a directly written Nb-C superconductor with a close-to-perfect edge barrier. The spatial evolution of the FFI is described using the edge-controlled FFI model, implying a chain of FFI nucleation points along the sample edge and their development into self-organized Josephson-like junctions (vortex rivers). In addition, our results offer insights into the applicability of widely used FFI models and suggest Nb-C to be a good candidate material for fast single-photon detectors.

[1] Faculty of Physics, University of Vienna, Boltzmanngasse 5, 1090 Vienna, Austria. [2] School of Physics, V. Karazin Kharkiv National University, Svobody Sq. 4, Kharkiv 61022, Ukraine. [3] Institute for Physics of Microstructures, Russian Academy of Sciences, Academicheskaya Str. 7, Afonino, Nizhny Novgorod region 603087, Russia. [4] Physics Department, Moscow Pedagogical State University, Malaya Pirogovskaya Str. 29/7, Bld. 1, Moscow 119435, Russia. [5] Institute of Physics, Goethe University, Max-von-Laue-Str. 1, 60438 Frankfurt, Germany. [6] B. Verkin Institute for Low Temperature Physics and Engineering of the National Academy of Sciences of Ukraine, Nauky Avenue 47, Kharkiv 61103, Ukraine. ✉email: oleksandr.dobrovolskiy@univie.ac.at

The dynamics of vortices at large transport currents is of major importance for the comprehension of vortex matter under far-from-equilibrium conditions and it sets practical limits for the use of superconductors in various applications[1–9]. The physics of current-driven vortex matter is getting especially interesting when the vortex velocity exceeds the velocity $v \approx 3\,\mathrm{km\,s^{-1}}$ of other possible excitations in the system, allowing for the Cherenkov-like generation of sound[10,11] and spin[12,13] waves by moving fluxons, which opens up novel routes to excite waves in magnon spintronics[14,15]. Furthermore, there is currently great interest in the interplay of Meissner currents and magnetic flux quanta with spin waves in the rapidly developing domain of magnon fluxonics[16,17].

The maximal current a superconductor can carry without dissipation is determined by the pair-breaking (depairing) current $I_{\mathrm{dep}}$. However, a highly resistive state in real systems is usually attained at much smaller currents due to the presence of regions in which superconductivity breaks down long before $I_{\mathrm{dep}}$ is reached. Namely, in a vortex-free state, the earlier breakdown of superconductivity is due to spatial variations of the order parameter caused by structural imperfectnesses and the sample geometry[18,19]. In the vortex state, fast-moving vortices are known to lead to a quench of the low-dissipative state at $I^* \ll I_{\mathrm{dep}}$ as a consequence of the flux-flow instability (FFI) associated with the escape of quasiparticles (normal electrons) from the vortex cores[20,21]. Accordingly, to achieve $I_c \lesssim I_{\mathrm{dep}}$ and high vortex velocities $v \gtrsim 5\,\mathrm{km\,s^{-1}}$, a high structural homogeneity and fast cooling of quasiparticles (governed by the quasiparticles' energy relaxation time $\tau_\epsilon$ and the escape time of nonequilibrium phonons to the substrate $\tau_{\mathrm{esc}}$) are both required. However, while short $\tau_\epsilon$ is inherent to disordered superconducting systems[22,23], few of them have $I_c \lesssim I_{\mathrm{dep}}$ in conjunction with weak volume pinning needed to maintain long-range order in the fast-moving vortex lattice. Variation in the local pinning forces induced by uncorrelated disorder (volume pinning) leads to a broader distribution of $v$ and thereby prevents the exploration of vortex matter at high velocities[24–27].

Recently, two approaches were used to demonstrate ultra-fast vortex motion at $v \gtrsim 5\,\mathrm{km\,s^{-1}}$. In the first case, a clean Pb bridge with both, an edge barrier for vortex entry and a high demagnetization factor (so-called geometrical barrier) was studied[6]. In the used geometry there was a strongly nonuniform current distribution both across and along the bridge due to a small Pearl length $2\lambda^2/d \ll w$, where $d$ and $w$ are the film thickness and width, respectively. A weak pinning and a short electron–phonon relaxation time $\tau_{\mathrm{ep}}$ in Pb[28] allowed one to diminish nonequilibrium effects and achieve the regime with ultra-fast Abrikosov–Josephson vortices[6]. In the second case, an array of ferromagnetic Co nanostripes on top of a superconducting Nb film led to a dynamic ordering of flux quanta guided by the nanostripes and allowed to achieve a narrow distribution of their velocities[29]. In both of these approaches, specially designed, locally nonuniform structures were used. At the same time, a close-to-ideal uniform system where the fast heat removal from electrons rather than the finite width of the $v$ distribution becomes the limiting factor for ultra-fast vortex dynamics was never investigated experimentally. Theoretically, however, it was recently predicted that dirty superconductors with weak volume pinning and strong edge barrier for vortex entry should also allow for ultra-fast vortex dynamics[30]. Extremely dirty superconductors are known to have a short electron–electron inelastic scattering time $\tau_{\mathrm{ee}}$ which leads to a decrease of $\tau_{\mathrm{ep}}$[31]. This diminishes nonequilibrium effects and may lead to an increase of the critical velocity of vortices. One of the most important requirements for the observation of an edge-controlled FFI is a spatially homogenous edge in conjunction with a weak pinning in the

superconductor's volume[30]. The presence of a strong edge barrier in such superconductors leads to a current gradient near the edge where vortices enter the superconductor and where FFI is actually nucleating.

Here, we demonstrate experimentally the phenomenon of edge-barrier-controlled FFI in direct-write superconductors with a close-to-perfect edge barrier and deduce vortex velocities up to $15\,\mathrm{km\,s^{-1}}$ from their current–voltage curves ($I$–$V$). The investigated system is the recently synthesized Nb-C superconductor fabricated by focused ion beam induced deposition (FIBID)[32], with a very high resistivity $\rho = 572\,\mu\Omega\mathrm{cm}$. This implies a large effect of the inelastic electron–electron scattering with the characteristic times $\tau_{\mathrm{ee}} \lesssim \tau_{\mathrm{ep}}$ which speeds up the relaxation of disequilibrium. The Nb-C microstrips have a rather low depinning current and their critical current is controlled by the edge barrier for vortex entry. In contrast to ref. [6], in our system $\lambda^2/d \gg w$, which means a negligible demagnetization factor (no geometrical barrier) and a uniform current distribution across the strip at zero magnetic field. The spatial evolution of the FFI is described in terms of the edge-barrier-controlled FFI model recently developed by one of the authors[30], implying a chain of FFI nucleation points along the sample edge and their development into self-organized Josephson-like junctions (vortex rivers) evolving to normal domains which expand along the entire sample. In addition, our results offer insights into the applicability of widely used FFI models and render Nb-C to be a good candidate material for fast single-photon detectors.

## Results

**System under investigations.** We study the vortex dynamics in a direct-write Nb-C superconducting microstrip fabricated by FIBID[32]. The microstrip is characterized by a transition temperature of $T_c = 5.6\,\mathrm{K}$ and close-to-depairing values of the zero-field critical current $I_c \approx 0.7 - 0.74 I_{\mathrm{dep}}$ above $0.5T_c$. The dimensions of the microstrip are: thickness $d = 15\,\mathrm{nm}$, width $w = 1\,\mu\mathrm{m}$, and length $l = 6.6\,\mu\mathrm{m}$, see Fig. 1 for the geometry. The microstrip is characterized by the coherence length at zero temperature $\xi(0) \approx 6.5\,\mathrm{nm}$, the penetration depth $\lambda(0) \approx 1060\,\mathrm{nm}$, and the Pearl length $2\lambda^2(0)/d \approx 150\,\mu\mathrm{m}$, that is $2\lambda^2(0)/d \gg w \gg \xi(0)$. The

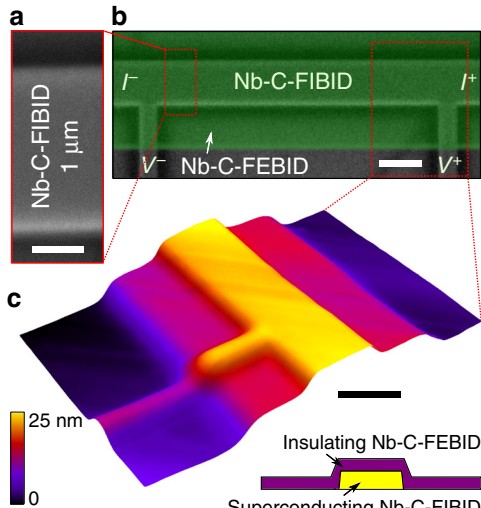

**Fig. 1 Experimental geometry.** Scanning electron microscopy images of the superconducting Nb-C-FIBID microstrip before (**a**) (the scale bar is 300 nm) and after (**b**) (the scale bar is 1 μm) covering it with an insulating Nb-C-FEBID layer shown by the green false color. The current and voltage leads are indicated with $I^+$, $I^-$, $V^+$, and $V^-$. **c** Atomic force microscopy image of a part of the fabricated structure. The scale bar is 1 μm.

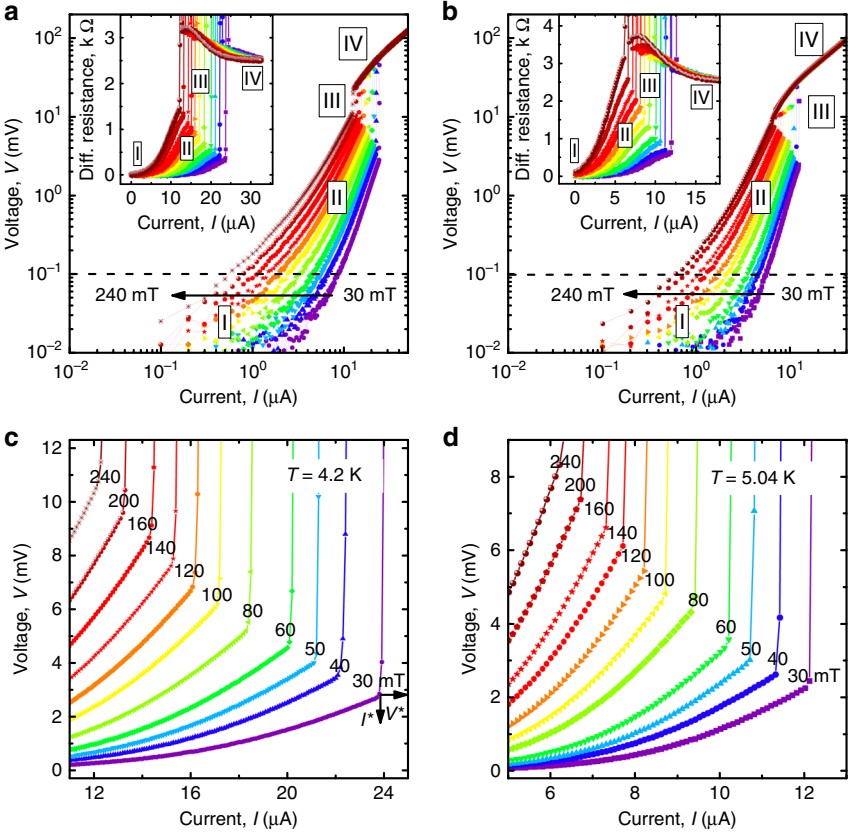

**Fig. 2 Current-voltage curves of the Nb-C-FIBID microstrip. a, b** $I$–$V$ curves of the microstrip in a series of magnetic fields at temperatures as indicated in panels (**c, d**). The different resistive regimes are indicated: pinned vortices (I), nonlinear conductivity in the flux-flow regime (II), flux-flow instability (III), and the normal state (IV). The instability jumps are enlarged in panels (**c, d**). The arrows in **c** illustrate the definition of the instability current $I^*$ related to the instability voltage $V^*$. Source data are provided as a Source Data file.

perpendicular-to-film-plane magnetic field with induction $\mathbf{B} = \mu_0\mathbf{H}$ populates the microstrip with a lattice of Abrikosov vortices. The applied dc current exerts a Lorentz force on the vortices that causes their motion with velocity $v$ across the microstrip. The associated voltage drop $V$ along the microstrip is recorded as a function of the applied current $I$ in the current-biased mode. The microstrip is capped with an insulating Nb-C layer fabricated by focused electron beam induced deposition (FEBID)[32,33]. Further details on the sample fabrication and its structural properties are given in the "Methods" section.

**Current-voltage characteristics.** Figure 2 displays the $I$–$V$ curves measured at 4.2 K ($0.75T_c$) and 5.04 K ($0.9T_c$) for a series of magnetic fields between 30 and 240 mT. With increase of the current, a series of different resistive regimes can be identified, as indicated in the $I$–$V$ curves: the pinned regime (I), the nonlinear flux-flow regime (II), and the FFI (III) causing abrupt onsets of the normal state (IV). Of especial interest for the following is the regime of high vortex velocities just before the FFI (III) with the $I$–$V$ sections enlarged in Fig. 2c, d.

From the last points before the voltage jumps, referring to Fig. 2c, d, the vortex instability velocity $v^*$ is deduced by the relation $v^* = V^*/(BL)$. The resulting dependence $v^*(B)$ is presented in Fig. 3a. Remarkably, $v^*$ is between 5 and 10 km s$^{-1}$ at larger fields $B \gtrsim 100$ mT and it is between 10 and 15 km s$^{-1}$ at $B < 100$ mT. The temperature dependence $v^*(T)$ is presented in Fig. 3b for two magnetic field values. The field 50 mT is exemplary for a relatively sparse vortex lattice (vortex lattice spacing $a \approx 220$ nm) while $a \approx 110$ nm at 200 mT for the assumed triangular vortex

lattice with $a = \sqrt{2\Phi_0/\sqrt{3}H}$, where $\Phi_0$ is the magnetic flux quantum. At both fields, the experimental data nicely fit the law $v^* \sim (1-t)^{1/4}$, where $t = T/T_c$, with $v^*(0.6T_c, 50$ mT$) = 12$ km s$^{-1}$ and $v^*(0.6T_c, 200$ mT$) = 7.7$ km s$^{-1}$, while a deviation of $v^*(B)$ from the $B^{-1/2}$ dependence is observed at $B \lesssim 50$ mT in Fig. 3a. The decreasing dependence of $v^*(B)$ below about 10 mT due to the decreasing vortex density (the so-called low-field crossover in the $v^*(B)$ dependence[34]) is beyond our consideration, as we are especially interested in the regime of very high vortex velocities.

**Influence of the edge barrier on the vortex dynamics.** The magnetic field dependence of the critical current at 4.20 K is presented in Fig. 3c. At smaller fields, $I_c(B)$ decreases linearly with $B$, while at larger fields the decrease of $I_c$ becomes nonlinear and slower. This behavior can be explained by the presence of some threshold field $B_{stop}$, which demarcates the Meissner (vortex free) and the mixed states of a superconducting stripe[35]. Namely, the dependence $I_c(B)$ in the Meissner state ($B < B_{stop}$) is linear and it is described by the expression $I_c(B) = I_c(0$ T$)(1 - B/2B_{stop})$, where $B_{stop}$ in the Ginzburg–Landau model[36] is given by $B_{stop} = B_s/2 = \Phi_0/(2\sqrt{3}\pi\xi(T)w)$. Here, $B_s$ is the field value at which the surface barrier for vortex entry is suppressed at $I = 0$, $\xi$ is the superconducting coherence length, and $w$ is the microstrip width. The definition of $B_{stop}$ following from $I_c(2B_{stop}) = 0$ is illustrated in Fig. 3c. For 10 mT $\lesssim B \lesssim 100$ mT, the dependence of the critical current is described well by the dependence $I_c(B) = I_c(0$ T$)B_{stop}/2B$, and $I_c(B)$ exhibits a linear decrease at low fields. At larger fields, $B \gtrsim 100$ mT, a further crossover at $B^*$ to a slower decrease of $I_c(B)$ as $B^{-0.5}$ is observed. The totality of our

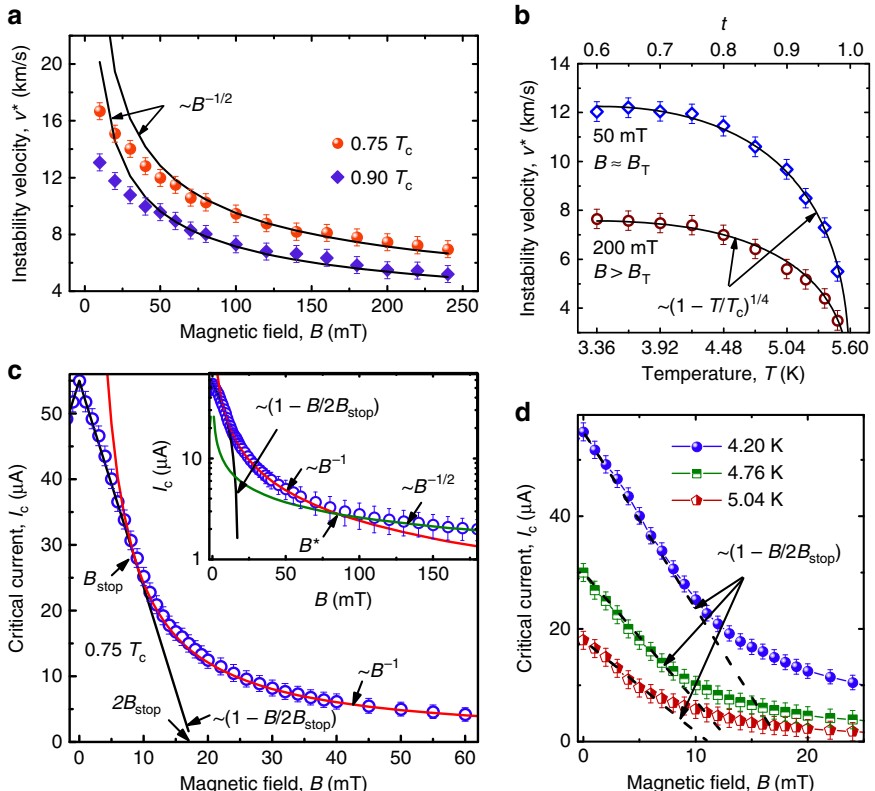

**Fig. 3 Instability velocity and critical current in the microstrip. a** Instability velocity $v^*$ as a function of the magnetic field. Red spheres and blue diamonds: experiment. Solid lines: fits to Eq. (2). **b** Temperature dependence of the instability velocity at 50 and 200 mT. Blue diamonds and brown circles: experiment. Solid lines: fits to Eq. (1). **c** Crossover from the linear dependence $I_c(B)$ at $B < B_{stop}$ to $I_c(B) \sim 1/B$ for $B_{stop} < B < B^*$ and $I_c(B) \sim 1/\sqrt{B}$ for $B > B^*$ at 4.20 K. Blue spheres: experiment. Solid lines: fits as labeled close to the curves. The inset shows the same data in $\log(I_c)$ versus $B$ representation. **d** Dependence of the critical current $I_c$ of the microstrip on the magnetic field at three different temperatures, as indicated. Symbols: experiment. Black dashed lines: linear fits. The error bars are the standard error of the mean.

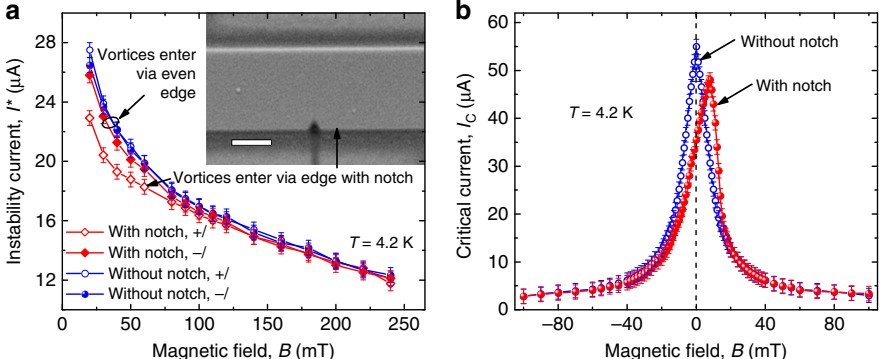

**Fig. 4 Edge-barrier effects on the flux-flow instability. a** Magnetic field dependence of the instability current $I^*(B)$ for a microstrip with even edges and a microstrip with an edge defect (notch) for two current polarities, as indicated. Inset: scanning electron microscopy image of the microstrip with the notch. The scale bar is 500 nm. **b** Magnetic field dependence of the critical current $I_c(B)$ for the two microstrips at 4.2 K. The error bars are the standard error of the mean. Source data are provided as a Source Data file.

experimental data indicates the dominating role of the edge mechanism[37] of vortex pinning in the studied sample at $B \lesssim 100$ mT, as is further commented in Supplementary Note 1.

**Influence of an edge defect on the vortex dynamics.** An additional reference measurement has been made for a microstrip with an artificially fabricated edge defect. The defect (notch) was milled by focused Ga ion beam at one edge of the microstrip and it has a shape of an equilateral triangle with a side of about 100

nm, see the inset in Fig. 4a. For a direct comparison of the edge-barrier effects on the vortex entry from different sides of the microstrip, the $I_c(B)$ and $I^*(B)$ curves are presented for both field and current polarities in Fig. 4. For the microstrip with even edges, the $I^*(\pm B)$ curves fall onto one another in the entire range of magnetic fields in Fig. 4a and the $I_c(B)$ dependence in Fig. 4b is symmetric with respect to the $B$ reversal. By contrast, for the microstrip with the notch, the maximum in $I_c(B)$ in Fig. 4b is shifted to $+8$ mT, in agreement with previous experiments on microstrips with defects (holes) close to one of their edges[38]. At

negative fields, the notch locally suppresses the edge barrier and thereby facilitates the entry of (anti)vortices. This leads to a small reduction of $I_c(B)$ up to larger field magnitudes at which the role of the volume pinning increases. At positive fields, when vortices enter the microstrip from the opposite side, the notch does not affect the vortex entry and this is why $I_c$ is not affected by the presence of the notch at $B \gtrsim 15$ mT. Remarkably, when vortices enter the microstrip via the edge with the notch, $I^*(B)$ at 20 mT $\lesssim B \lesssim 100$ mT decreases by up to about 10% in comparison with $I^*(B)$ when vortices enter from the opposite side, which is in line with the calculations[39]. Importantly, due to the nonlinear upturns of the $I$–$V$ curves just before the instability jump, a decrease of $I^*$ by about 10% leads to a stronger decrease of the instability velocity $v^*$. This provides a direct evidence of the decisive role of the edge barrier on the FFI, as will be detailed next.

## Discussion

We first compare the experimental results with the widely used Larkin–Ovchinnikov (LO) FFI model[21,40] with the modifications introduced by Bezuglyj and Shklovskij (BS)[41] and Doettinger et al.[42]. Although edge-barrier effects are not considered in these models[21,40–42], it is still interesting to check what quasiparticle energy relaxation time $\tau_\epsilon$ values, related to the instability velocity, can be deduced from fitting of the experimental data to these models.

Within the framework of the LO theory[21,40], the microscopic mechanism of FFI is the following. When the electric field induced by vortex motion raises the quasiparticle energy above the potential barrier associated with the order parameter around the vortex core, quasiparticles leave it and the core shrinks. The shrinkage of the vortex cores leads to a reduction of the viscous drag coefficient and a further avalanche-like acceleration of the vortex, eventually quenching the low-resistive state. The original LO theory was developed in the dirty limit near $T_c$ and in neglect of heating of the superconductor. To account for quasiparticle heating due to the finite heat-removal rate of the power dissipated in the sample, the LO theory was extended by BS[41]. In the BS generalization, the latter effect was considered in the framework of the kinetic equation LO approach, which assumes a non-thermal (non-Fermi–Dirac) electron distribution function, while Joule heating was taken into account using a thermal distribution function and the electron temperature $T_e$ was determined from

the heat conductance equation. In contrast to the $B$-independent instability velocity $v^*$ in the LO model, a $v^*(B)$ variation is expected in the BS model[41] and takes the form:

$$v^* \propto h(1-t)^{1/4}B^{-1/2}, \tag{1}$$

where $h$ is the heat removal coefficient. While the magnetic field dependence $v^*(B)$ nicely fits Eq. (1) at $B \gtrsim 50$ mT, a notable deviation of $v^*(B)$ toward smaller values is observed in Fig. 3a at $B \lesssim 50$ mT. This deviation will be commented in what follows. In all, the complete set of the instability parameters deduced from Fig. 2 nicely fits the BS scaling law, see Supplementary Fig. 1. However, if one associates $\tau_\epsilon$ with the electron–phonon scattering time $\tau_{ep}$ in the LO model, the deduced $\tau_\epsilon$ is at least one order of magnitude smaller than one could expect from $\tau_\epsilon$ found in similar low-$T_c$ highly disordered superconductors[43–45], see Supplementary Discussion.

In the LO model modified by Doettinger et al.[42,46], the quasiparticle energy relaxation time can be found from the following equation:

$$v^* = \left[\frac{(1-t)^{1/2}D[14\zeta(3)]^{1/2}}{\pi\tau_\epsilon}\right]^{1/2}\left(1 + \frac{a}{\sqrt{D\tau_\epsilon}}\right). \tag{2}$$

In Eq. (2), the term $a/\sqrt{D\tau_\epsilon}$, where $a$ is the intervortex distance, has been added to incorporate the necessary condition of spatial homogeneity of the nonequilibrium quasiparticle distribution between vortices at relatively small magnetic fields. The calculation results by Eq. (2) are shown by solid lines in Fig. 3a where the energy relaxation time has been varied as the only fitting parameter. The best fits are achieved with $\tau_\epsilon = 16$ ps which could be considered as a more accurate estimate for the energy relaxation time in the Nb-C-FIBID superconductor. We note that with this $\tau_\epsilon$ estimate, the quasiparticle diffusion length $l_\epsilon = \sqrt{D\tau_\epsilon} \approx 28$ nm is much smaller than the intervortex distance $a$ at all used magnetic fields and, importantly, $l_\epsilon \lesssim 2\xi(T)$ with $2\xi$ $(0.75T_c) \approx 25$ nm and $2\xi(0.9T_c) \approx 38$ nm.

The edge-barrier-controlled FFI scenario[30] is different from the FFI scenario of LO and BS. Indeed, LO and BS considered a moving periodic vortex lattice in an infinite superconductor in the Wigner–Seitz approximation and hence could not take into account the collective effects related to the transformation of the vortex lattice and edge-barrier effects. In contrast, in the edge-

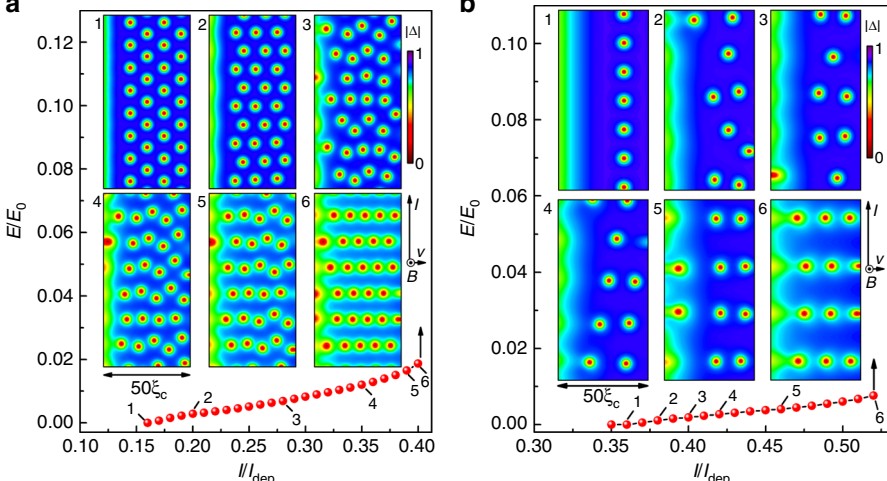

**Fig. 5 Time-dependent Ginzburg–Landau simulations.** Calculated $I$–$V$ curves of a superconducting microstrip with width $w = 50\xi_c$ at $T = 0.8T_c$ for $B = 0.02\,B_0$ (**a**) and $B = 0.05\,B_0$ (**b**). The insets show snapshots of the superconducting order parameter $|\Delta|$ at different current values, as indicated. For the studied system, the parameter $B_0 = \Phi_0/(2\pi\xi_c^2) \simeq 4.9$ T, where $\xi_c = \sqrt{1.76}\xi(0) = 8.2$ nm. The electric field is measured in units of $E_0 = k_BT_c/(2e\xi_c)$ and the current in units of $I_{dep}$.

barrier-controlled FFI model[30] a nonuniform distribution of vortices is taken into account, as well as the local Joule heating and cooling (due to the time variation of the magnitude of the superconducting order parameter $|\Delta|$) depending on the vortex position. The edge-barrier-controlled FFI model allows for studying a "local" instability and collective effects in the vortex dynamics relying upon the solution of a heat conductance equation for the electrons and a modified time-dependent Ginzburg–Landau equation for $\Delta(r, t)$. In this model, it was shown that, in the low-resistive state, there is a temperature gradient across the width of the microstrip with maximal local temperature near the edge where vortices enter the sample[30]. The higher temperature at the edge is caused by the larger current density in the near-edge area due to the presence of the edge barrier for vortex entry and, hence, the locally larger Joule dissipation. With increase of the current, there is a series of transformations of the moving vortex lattice. In Fig. 5, we show examples of the calculated $I$–$V$ curves and snapshots of $|\Delta|(r)$ for the parameters of the superconductor as in ref. [30]. Similar transformations connected with reorientations of the moving vortex lattice in the insets 1–2 in Fig. 5b were experimentally observed[47] and theoretically analyzed[48] previously.

At currents just below $I^*$, localized areas with strongly suppressed superconductivity and closely spaced vortices appear near the hottest edge (left edge in the insets in Fig. 5). Upon reaching $I^*$, these areas begin to grow in the direction of the opposite edge and form a highly resistive Josephson SNS-like link (vortex river) along which vortices move[3,6,30,49]. These vortices are of the Abrikosov–Josephson type, as they are moving in areas with suppressed order parameter. Due to the increasing dissipation, vortex rivers evolve into normal domains which than expand along the microstrip. In consequence of this, a jump to the highly resistive state occurs at $I^*$. In all, the simulation results demonstrate that transformation of the moving vortex array is a collective phenomenon, which involves correlated changes in the motion of many vortices with increase of the current and, at $I^*$, results in the appearance of Josephson-like SNS links known as vortex rivers[3,6,49].

In the edge-controlled FFI model[30], the current $I^*$ increases linearly with the width of the strip, while $V^*$ does not depend on $w$ as it does in the LO and BS models. This result holds at $B \gg B_{stop}$ when $a$ is much smaller than the microstrip width $w$ and $a$ becomes smaller than the width of the vortex-free region near the edge of the microstrip. This means that despite the nucleation of FFI points occurs near the edge where the local temperature and the current densities are maximal, far from the edge where current density is uniform, the vortices should move at relatively high velocities. Otherwise the FFI will not develop across the whole microstrip and one has only origins of the vortex rivers, as it can be seen from Fig. 5 in[30] at $I \lesssim I^*$. The linear scaling of $I^*(w)$ with the microstrip width $w$ is corroborated by the experimental observation in Fig. 6a, where the $I$–$V$ curves for two microstrips with the widths $w = 1$ μm and 500 nm are shown at $T = 4.2$ K and $B = 50$ mT.

In the edge-barrier-controlled FFI model[30], the energy relaxation time depends not only on the electron–phonon relaxation time $\tau_{ep}$ (as in the LO model) but also on the escape time of nonequilibrium phonons to the substrate $\tau_{esc}$ and the ratio of the electron and phonon heat capacities, $C_e$ and $C_p$, respectively. At $T \simeq T_c$ and for a small deviation from equilibrium one has:

$$\tau_\epsilon \simeq \tau_E + \tau_{esc}(1 + C_e(T_c)/C_p(T_c)), \qquad (3)$$

where $\tau_E \simeq \tau_{ep}/4.5$ is the electron–phonon relaxation time renormalized due to fast electron–electron inelastic scattering. Here, $\tau_{ep}$ is the electron–phonon relaxation time used in the LO model. Following the arguments of ref. [42], one can claim that the

instability occurs at the velocity $v^* \sim a/\tau_\epsilon$ when the intervortex distance is $a \lesssim \sqrt{D\tau_\epsilon}$. This condition leads to a dependence of $v^*(B)$, which was revealed in numerical calculations[30]. One important difference between the modified LO model[42] and the edge-controlled FFI model is that in the latter[30], $a \sim B^{-1/2}$ only at relatively large magnetic fields, when the intervortex distance at $I \sim I_c$ and $I \sim I^*$ is almost the same despite the change in the structure of the moving vortex lattice. At relatively small magnetic fields, $a$ in the vortex rows is smaller than $(2\Phi_0/B\sqrt{3})^{1/2}$ at $I \sim I^*$ and, thus, the number of vortices is smaller than follows from the simple estimate $n\Phi_0 = BS$, see Fig. 5a. Altogether, this leads to a weaker experimental dependence $v^*(B)$ than follows from the "global" instability model with $v^* \sim B^{-1/2}$[42]. Qualitatively, it is this behavior which is observed in the experiment, see Fig. 3a.

The large $v^*$ values observed in our system should be attributed not only to $\tau_E < \tau_{ep}$ but, also, to a small $\tau_{esc}$ in Eq. (3). Indeed, due to the insulating Nb-C-FEBID layer on top of the microstrip, there seems to be no phonon bottleneck which could exist due to an acoustic mismatch between a thin dirty superconductor and a substrate[44]. As an estimate, for our system we deduce $\tau_{esc} \sim 4d/u \approx 24$ ps, where $u \sim 2.5$ km s$^{-1}$ is the mean sound velocity. This value is larger than $\tau_\varepsilon \sim 16$ ps deduced from the experimental data using the modified LO model. We have to stress that numerical coefficients in the LO model are strictly valid only rather close to $T_c$ (when $\Delta(T) \ll k_B T_c$, i.e., at $T \gtrsim 0.9 T_c$) and in the case when $\tau_{ee} \gg \tau_{ep}$ and $\tau_{esc} = 0$. Therefore these coefficients may be different in our dirty system with $\tau_\epsilon \sim \tau_{esc}$ and at temperatures further away from $T_c$.

Finally, we would like to note that, unfortunately, there is no analytical relation between $v^*$ and $\tau_\epsilon$ in the edge-barrier-controlled FFI model[30]. Accordingly, a discussion of the relation between $v^*$ and $\tau_\epsilon$ has to remain on a qualitative level. From Eq. (3) it follows that a change of $\tau_E$, $\tau_{esc}$, and $C_e/C_p$ leads to a change of the relaxation time $\tau_\epsilon$. To illustrate this, in Fig. 6b we present a series of calculated $I$–$V$ curves at different $\tau_{esc}$ values, while the other parameters are kept fixed. Indeed, with increasing $\tau_{esc}$ the critical velocity $v^* \sim E^*$ decreases, but it decreases slower than $\tau_\epsilon^{-1}$ or $\tau_\epsilon^{-1/2}$. Qualitatively, the same tendency is found if one increases the ratio $C_e/C_p$ for a given $\tau_{esc}$ value. Specifically, with an increase of $\tau_{esc}/\tau_E$ by two orders of magnitude, $E/E_0$ decreases by only about a factor of three. In the inset of Fig. 6b, one can also see that with the increase of $\tau_{esc}$, the time-averaged temperature in the center of the superconducting microstrip increases, which indicates an increased contribution of Joule dissipation to the FFI. The increased temperature also affects $v^*$ because of the temperature dependence $\tau_E \sim 1/T^3$ and $C_e/C_p \sim 1/T^2$ in the used model[30].

We would like to outline an applications-related aspect of the superconducting properties of the studied Nb-C-FIBID microstrip. Namely, the small diffusivity $D \approx 0.49$ cm$^2$ s$^{-1}$ and the low transition temperature $T_c = 5.6$ K suggest that Nb-C-FIBID may be a candidate material for superconducting single-photon detectors (SSPDs). We refer to Table 1 for a comparison with parameters of some typical SSPDs and to ref. [31] for a further discussion. In this regard, it should be mentioned that for about a decade SSPDs were made of meandering nanostrips with widths in the range 50–150 nm as it was empirically found that the use of wider strips leads either to the loss of the single-photon nature of the response or to a decrease of the detection efficiency[50]. This observation was in line with a "geometric-hot-spot" detection model, in which the width of the supercurrent-carrying strip should be comparable with the diameter of the normal region where the superconducting state is suppressed due to the absorption of the photon.

Recently, a "photon-generated superconducting vortex model" was proposed[31,51]. It was revealed that the efficiency of the photon

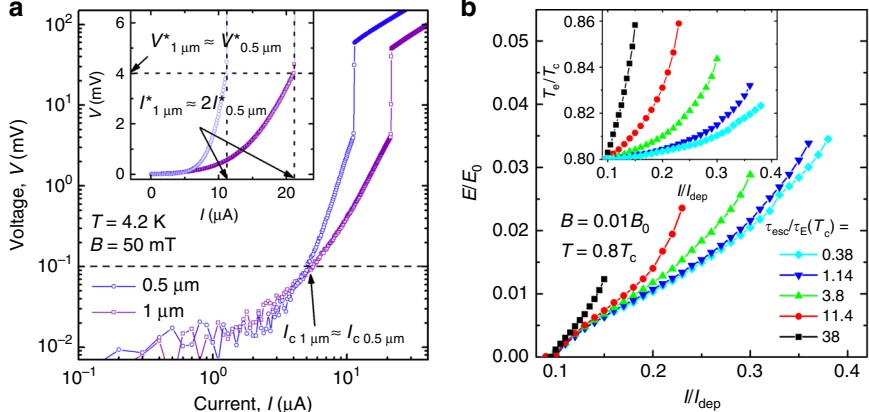

**Fig. 6 Current–voltage curves of the microstrips. a** Experimental $I$–$V$ curves of the two Nb-C-FIBID microstrips with the widths $w = 1\,\mu m$ and 500 nm at $T = 4.2\,K$ and $B = 50\,mT$ in the double log representation. Inset: the same data in the linear scale. Source data are provided as a Source Data file. **b** Calculated $I$–$V$ curves of a superconducting microstrip with the width $w = 50\xi_c$ at $T = 0.8T_c$, $B = 0.01B_0$ for different $\tau_{esc}$ values, as indicated, for $C_e(T_c)/C_p(T_c) = 0.57$, $\tau_E = 12.5\,ps$, and $\tau_E(0.8T_c) \simeq 2\tau_E(T_c)$. Inset: time-averaged electronic temperature $T_e$ in the center of the microstrip as a function of the normalized current.

---

**Table 1 Nb-C-FIBID as a candidate material for single-photon detectors.**

| Material | MoSi[53] | NbRe[69] | NbN[52] | NbN[70] | Nb-C[52] | Nb-C-FIBID |
|---|---|---|---|---|---|---|
| $d$, nm | 3.3 | 15 | 5.8 | 14.4 | 23.3 | 15 |
| $T_c$, K | 3.85 | 6.77 | 8.35 | 15.25 | 11.2 | 5.6 |
| $\rho_n$, $\mu\Omega$cm | 175 | 145 | 400 | 281 | 25 | 572 |
| $D$, cm$^2$s$^{-1}$ | 0.47 | 0.56 | 0.31 | 0.6 | 4.45 | 0.49 |
| $\lambda(0)$, nm | 708 | 483 | 450[a] | 290[a] | 156[a] | 1060 |
| $\xi(0)$, nm | 8.7 | 4.8 | 5.4 | 5.4 | na | 6.5 |
| $I_c/I_{dep}$ | $\simeq 0.7$ | na | na | 0.65–0.9 | na | 0.7–0.74 |
| Leads | Tapered | Straight | Tapered | Straight | Straight | Straight |

$d$: stripe thickness; $T_c$: superconducting transition temperature; $\rho_n$: resistivity just above $T_c$; $\lambda(0)$: estimate for the penetration depth at zero temperature; $\xi(0)$: zero-temperature estimate for the coherence length.
*na:* not available.
[a]An estimate which was made on the basis of the reported data.

---

detection is not determined by the geometry, as long as the initial current density is uniform and close to the critical pair-breaking current $I_{dep}$. It was emphasized that even several micron wide dirty superconducting stripes should be suitable to detect single near-infrared or optical photons if their critical current $I_c \gtrsim 0.7I_{dep}$[31]. The only requirement for the width of the strip is that it should be smaller than the Pearl length $\Lambda = 2\lambda^2/d$ that ensures the uniformity of the supercurrent across the superconductor width. Recently, this condition was satisfied in wide and short NbN[52] and MoSi[53] bridges, whose photon response was consistent with the vortex-assisted mechanism of initial dissipation[51]. In this way, given the superconducting properties of our samples, which drastically differ from much cleaner Nb-C films prepared by pulsed laser ablation in ref. [54], Nb-C-FIBID appears to be a good candidate for fast single-photon detection. A further enhancement of the critical current in Nb-C-FIBID can be expected for tapered current leads[52,53] which should minimize the reduction of $I_c$ in consequence of undesired current-crowding effects[19], and additional advantages of easy on-chip[55] or on-fiber[56] integration are provided by the direct-write nanofabrication technology. Furthermore, the ability to control the thickness of individual FIBID/FEBID layers with an accuracy better than 1 nm[57,58] should allow for the fabrication of superconductor/insulator superlattices for studying quantum interference and commensurability effects[59] as well as photonic crystals with superconducting layers[60].

To summarize, we have experimentally demonstrated ultra-fast vortex dynamics at velocities up to $15\,km\,s^{-1}$ in a uniform superconducting microstrip fabricated by FIBID. A stable flux flow at such high velocities is a consequence of the combined effects of a strong edge barrier against a background of weak volume pinning, close-to-depairing critical currents, and fast quasiparticles relaxation in the investigated system. The distinctive feature of the direct-write Nb-C superconductor is a close-to-perfect edge barrier which orders the vortex motion at large current values and allows for the description of the spatial evolution of the FFI relying upon the edge-barrier-controlled FFI model. The observed high vortex velocities in Nb-C-FIBID make accessible studies of far-from-equilibrium superconductivity[61] and vortex matter driven by large currents, opening prospects for Cherenkov-like generation of other excitations by the fast-moving vortex lattice in ferromagnet/superconductor hybrid structures. In addition, the small electron diffusion coefficient $D \approx 0.5\,cm^2\,s^{-1}$, the low superconducting transition temperature $T_c = 5.6\,K$, and high $I_c$ values exceeding 70% of the depairing current render Nb-C-FIBID to be an interesting candidate material for fast single-photon detectors.

## Methods

**Sample fabrication and its structural properties.** Superconducting microstrips were fabricated by FIBID in a dual-beam scanning electron microscope (FEI Nova Nanolab 600). The substrates are Si (100, p-doped)/SiO$_2$ (200 nm) with lithographically defined Au/Cr contacts for electrical transport measurements[62]. FIBID was done at 30 kV/10 pA, 30 nm pitch and 200 ns dwell time employing Nb (NMe$_2$)$_3$(N-$t$-Bu) as precursor gas. The as-deposited Nb-C-FIBID microstrips have well-defined smooth edges and an rms surface roughness of <0.3 nm, as deduced

from atomic force microscopy scans in the range $1 \times 1$ μm. Right after the deposition, without breaking the vacuum, the microstrips were covered with a 10-nm-thick insulating Nb-C layer prepared focused by FEBID[33,63], see Fig. 1 for the geometry. While Nb-C-FEBID structures are amorphous, Nb-C-FIBID deposits have an fcc Nb-C polycrystalline structure, with grains about 15 nm in diameter[32]. The typical elemental composition in the Nb-C-FIBID microstrips is 43% at. C, 29% at. Nb, 15% at. Ga, and 13% at. N, as inferred from energy-dispersive X-ray spectroscopy on thicker replica of the fabricated structures. Experiments were done on a series of four samples. In the manuscript, we report typical data for one microstrip. An additional reference measurement has been made for a microstrip with an artificially fabricated edge defect. The defect (notch) was milled by focused Ga ion beam at a beam voltage of 30 kV and a beam current of 10 pA[64].

**Superconducting properties of the Nb-C-FIBID microstrip**. The resistive properties of the microstrip are summarized in Fig. 7. The resistivity temperature dependence $\rho(T)$ is shown in Fig. 7a, where the $\rho(T)$ curve exhibits a transition from weak localization[65] to superconductivity at $T_c = 5.6$ K. Here, the transition temperature $T_c$ is determined using the 50% resistance drop criterion, as illustrated in Fig. 7b. The resistivity at 7 K is $\rho_{7K} = 572$ μΩcm and the width of the superconducting transition, defined as the temperature difference between the 10 and 90% resistivity values at the transition, amounts to $\Delta T_c \approx 0.6$ K. Application of a magnetic field $B$ leads to a decrease of $T_c$ and a transition broadening, and we use the same 50% resistance drop criterion to deduce the temperature dependence of the upper critical field $B_{c2}(T)$ shown in Fig. 7c. Near $T_c$, the critical field slope $dB_{c2}/dT|_{T_c} = -2.24$ T K$^{-1}$ corresponds, in the dirty superconductor, to the electron diffusivity $D = -4k_B/[\pi e(dB_{c2}/dT|_{T_c})] \approx 0.49$ cm$^2$ s$^{-1}$. The coherence length and the penetration depth at zero temperature are estimated[52] as $\xi(0) = \sqrt{\hbar D/\Delta(0)} = 6.5$ nm and $\lambda(0) = 1.05 \cdot 10^{-3}\sqrt{\rho_{7K}/T_c} \approx 1060$ nm. By employing the 100 μV voltage drop criterion, from the $I$–$V$ curves, we deduce the critical currents at zero field $I_c(0.75T_c) = 58$ μA and $I_c(0.9T_c) = 16$ μA. We assume that the temperature dependence of the depairing current can be described by the expression $I_{dep}(T) = I_{dep}(0)(1 - (T/T_c)^2)^{3/2}$ with the prefactor $I_{dep}(0) = 0.74w[\Delta(0)]^{3/2}/(eR_\square\hbar D)$, which is justified for dirty superconductors[52,66,67]. Here, $\Delta(0)$ is the superconducting energy gap at zero temperature, $e$ the electron charge, and $R_\square$ the sheet resistance. With the assumed BCS ratio $\Delta(0) \approx 1.76k_BT_c$, we obtain $I_{dep}(0) \approx 268$ μA. The calculated dependence $I_{dep}(T)$ is compared with the

experimentally measured $I_c(T)$ in Fig. 7d. We note that $I_c$ varies between $0.7I_{dep} \lesssim I_c \lesssim 0.74I_{dep}$ in the temperature range $0.5 < t < 1$, where $\tau = T/T_c$ is the reduced temperature.

**Time-dependent Ginzburg–Landau simulations**. To study the evolution of the superconducting order parameter, we numerically solve the modified TDGL equation[31]:

$$\frac{\pi\hbar}{8k_BT_c}\left(\frac{\partial}{\partial t} + \frac{2ie\varphi}{\hbar}\right)\Delta = \xi_{mod}^2\left(\nabla - i\frac{2e}{\hbar c}A\right)^2\Delta + \left(1 - \frac{T_e}{T_c} - \frac{|\Delta|^2}{\Delta_{mod}^2}\right)\Delta +$$
$$+ i\frac{(\text{div}\mathbf{j}_s^{Us} - \text{div}\mathbf{j}_s^{GL})}{|\Delta|^2}\frac{e\Delta\hbar D}{\sigma_n\sqrt{2}\sqrt{1 + T_e/T_c}},$$

where $\xi_{mod}^2 = \pi\sqrt{2}\hbar D/(8k_BT_c\sqrt{1 + T_e/T_c})$, $\Delta_{mod}^2 = x(\Delta_0\tanh(1.74\sqrt{T_c/T_e-1}))^2/(1 - T_e/T_c)$, $A$ is the vector potential, $\varphi$ is the electrostatic potential, $D$ is the diffusion coefficient, $\sigma_n = 2e^2DN(0)$ is the normal-state conductivity with $N(0)$ being the single-spin density of states at the Fermi level, and $\mathbf{j}_s^{Us}$ and $\mathbf{j}_s^{GL}$ are the superconducting current densities in the Usadel and Ginzburg–Landau models:

$$\mathbf{j}_s^{Us} = \frac{\pi\sigma_n}{2e\hbar}|\Delta|\tanh\left(\frac{|\Delta|}{2k_BT_e}\right)\mathbf{q}_s,$$

where $\mathbf{q}_s = \nabla\phi - 2e\mathbf{A}/\hbar c$, $\phi$ is a phase of $\Delta = |\Delta|e^{i\phi}$, and $\mathbf{j}_s^{GL} = \frac{\pi\sigma_n|\Delta|^2}{4e\hbar k_BT_c}\mathbf{q}_s$. It should be noted that at $T_e$ not very close to $T_c$ the Ginzburg–Landau expression for the superconducting current is not valid quantitatively and one needs to use the Usadel expression for $\mathbf{j}_s^{Us}$. In this case, one should also modify the TDGL equation since the ordinary TDGL equation leads to $\text{div}\mathbf{j}_s^{GL} = 0$ in the stationary case, while one needs $\text{div}\mathbf{j}_s^{Us} = 0$. Accordingly, by adding the term $\text{div}(\mathbf{j}_s^{Us} - \mathbf{j}_s^{GL})$ in the TDGL equation we provide $\text{div}\mathbf{j}_s^{Us} = 0$. At $T_e \to T_c$ the modified TDGL equation reduces to the ordinary TDGL equation and $\text{div}(\mathbf{j}_s^{Us} - \mathbf{j}_s^{GL})$ goes to 0.

The electron and phonon temperatures, $T_e$ and $T_p$, respectively, are found from the solution of following equations:

$$\frac{\partial}{\partial t}\left(\frac{\pi^2k_B^2N(0)T_e^2}{3} - \mathcal{E}_0\mathcal{E}_s(T_e, |\Delta|)\right) = \nabla k_s\nabla T_e - \frac{96\zeta(5)N(0)k_B^5}{\tau_0}\frac{T_e^5 - T_p^5}{T_c^5} + jE,$$
$$\frac{\partial T_p^4}{\partial t} = -\frac{T_p^4 - T^4}{\tau_{esc}} + \gamma\frac{24\zeta(5)}{\tau_0}\frac{15}{\pi^4}\frac{T_e^5 - T_p^5}{T_c},$$

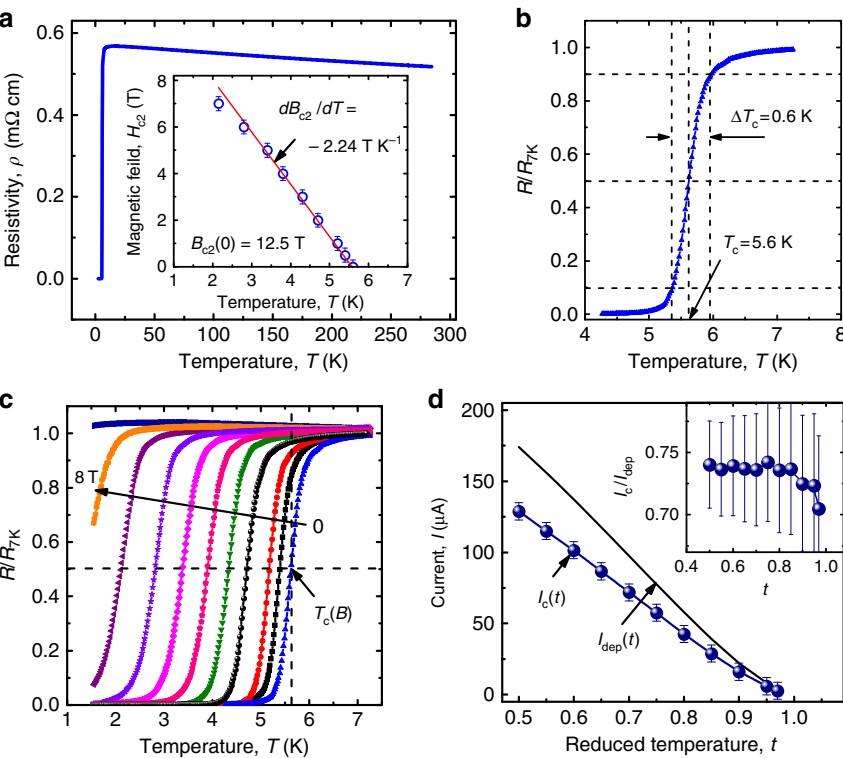

**Fig. 7 Superconducting properties of the Nb-C-FIBID microstrip. a** Temperature dependence of the resistivity of the microstrip. Inset: experimental temperature dependence of the upper critical field (blue circles) fitted to the expression $B_{c2}(T) = B_{c2}(0) - (dB_{c2}/dT)T$ with $B_{c2}(0) = 12.5$ T and $dB_{c2}/dT = -2.24$ T K$^{-1}$ (red solid line). **b** Transition to the superconducting state in zero magnetic field. **c** Evolution of the superconducting transition in the presence of a magnetic field. **d** Temperature dependence of the experimentally measured critical current $I_c(t)$ (blue spheres) in comparison with the theoretically calculated depairing current $I_{dep}(t)$ (black solid line). Inset: ratio $I_c/I_{dep}$ versus reduced temperature $t$. The error bars are the standard error of the mean. Source data are provided as a Source Data file.

where $\mathcal{E}_0 = 4N(0)(k_B T_c)^2$, $\mathcal{E}_0 \mathcal{E}_s(T_e, |\Delta|)$ is the change in the energy of electrons due to the transition to the superconducting state, $k_s$ is the heat conductivity in the superconducting state:

$$k_s = k_n \left( 1 - \frac{6}{\pi^2 (k_B T_e)^3} \int_0^{|\Delta|} \frac{\epsilon^2 e^{\epsilon/k_B T_e} d\epsilon}{(e^{\epsilon/k_B T_e} + 1)^2} \right),$$

$k_n = 2D\pi^2 k_B^2 N(0) T_e/3$ is the heat conductivity in the normal state, the term $jE$ describes Joule dissipation, and $\tau_{esc}$ is the escape time of nonequilibrium phonons to the substrate. The parameter $\gamma$ is defined as $\gamma = \frac{8\pi^2}{5} \frac{C_e(T_c)}{C_p(T_c)}$, where $C_e(T_c)$ and $C_p(T_c)$ are the heat capacities of electrons and phonons at $T = T_c$, and the characteristic time $\tau_0$ controls the strength of the electron–phonon and phonon–electron scattering[31]. It should be noted that the electron–photon scattering time enters the TDGL equation indirectly via the electron temperature $T_e$ whose dynamics is governed by $\tau_{e-ph} \sim \tau_0$ in the heat conductance equation. This is rather similar to the LO approach, where $\tau_{e-ph}$ enters the kinetic equation for the electron distribution function $f(E)$ (in our case this is the heat conductance equation for $T_e$) and $f(E)$ enters the GL equation in the LO model[20,21].

To find the electrostatic potential $\varphi$, we also solve the current continuity equation:

$$\text{div}(\mathbf{j}_s^{Us} + \mathbf{j}_n) = 0,$$

where $\mathbf{j}_n = -\sigma_n \nabla \varphi$ is the normal current density.

Values of the parameters $\gamma = 9$ and $\tau_0 = 925$ ns used in the calculations are estimates for NbN. Their variation only leads to quantitative changes in the $I$–$V$ curves.

At the edges where vortices enter and exit the microstrip, we use the boundary conditions $\mathbf{j}_n|_n = \mathbf{j}_s|_n = 0$ and $\partial T_e/\partial n = 0$, $\partial |\Delta|/\partial n = 0$ while at the edges along the current direction $T_e = T$, $|\Delta| = 0$, $\mathbf{j}_s|_n = 0$, $\mathbf{j}_n|_n = I/wd$. The latter boundary conditions model the contact of the superconducting strip with a normal reservoir being in equilibrium. This choice provides a way "to inject" the current into the superconducting microstrip in the modeling. The modeled length of the microstrip is $L = 4w$.

In the considered model, the penetration length of the electric field $L_E$ is about the coherence length $\xi(T)$, which is a consequence of $\tau_{ee} \ll \tau_{ep}$. If $\tau_{ee} \gtrsim \tau_{ep}$, then $L_E$ can be considerably larger than $\xi(T)$. In general, $L_E$ stipulates the stability of the phase slip process in 1D superconducting wires at larger currents[68]. In the case of vortex rivers (phase slip lines with vortices) it should also lead to their stability at larger currents, providing a critical velocity of Abrikosov vortices close to the velocity of Josephson vortices, which could explain the experimentally observed $v^* \gtrsim 10$ km s$^{-1}$. Within the framework of the considered model, a larger $L_E$ can be modeled by a smaller numerical coefficient at the time derivative $\partial \Delta/\partial t$. This simultaneously leads to a decrease of the relaxation time of $|\Delta|$, which also leads to an increase of $v^*$. For instance, a fivefold decrease of this coefficient (that corresponds to an increase of $L_E$ by a factor of $\approx \sqrt{5}$) results in a twofold increase of $V^*$ and $v^*$ and a small decrease of $I^*$ at $B = 0.1B_0$. One can also see that in this case vortex rivers are well formed at $I = I^*$ and Abrikosov vortices are closer to Abrikosov–Josephson vortices because of the stronger suppression of the order parameter along the vortex river, leading to higher instability velocities.

## Data availability

The authors declare that the data supporting the findings of this study are available within the paper and its supplementary information files. The source data underlying Figs. 2, 4a, 6, and 7 are provided as a Source Data file. Source data are provided with this paper.

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

## Acknowledgements

O.V.D. acknowledges the German Research Foundation (DFG) for support through Grant No. 374052683 (DO1511/3-1). D.Y.V. acknowledges the Russian Science Foundation for support through Grant No. 17-72-30036. A.V.C. acknowledges support within the ERC Starting Grant No. 678309 MagnonCircuits. Furthermore, this work was supported by the European Cooperation in Science and Technology via COST Action CA16218 (NANOCOHYBRI). Support through the Frankfurt Center of Electron Microscopy (FCEM) is gratefully acknowledged. Open access funding has been provided by the University of Vienna.

## Author contributions

O.V.D. conceived the experiment and performed the measurements. O.V.D. and M.Y.M. designed the samples. F.P. and R.S. fabricated the samples under the supervision of M.H. R.S. automated the data acquisition. O.V.D. and V.M.B. evaluated the data. D.Y.V. provided theoretical support and performed simulations. O.V.D., D.Y.V., A.V.C., and M.H. discussed the interpretation and the relevance of the results. O.V.D. and D.Y.V. wrote the paper. All authors discussed the results and contributed to the paper writing.

## Competing interests

The authors declare no competing interests.
