## [Peer Review File · Nature Communications]

Reviewers' comments:

Reviewer #1 (Remarks to the Author):

This work reports extensive transport measurements of the flux flow instability which was interpreted in terms of the Larkin-Ovchinnikov (LO) instability affected by the edge barrier. This study extends many previous experimental investigations of the LO instability to a narrow film Nb-C film strips with controlled edge barrier properties which allowed the authors to extract the vortex velocities of the order of 10 km/s at the onset of instability. The results appear interesting and qualitative conclusions are supported by the experimental data, so this work in my opinion would be appropriate for Nature Communications. Yet the authors focused on a particular scenario, while disregarding other effects which can affect the value of the parameters extracted from model fits, so the key assumptions should be clarified, as outlined below.

1. The field dependence $I_c(B) \propto B^{-a}$ with $a \approx 1/2$ observed at $B > 100$ mT has been often observed on superconducting films. It follows from many models of bulk pinning [see, e.g., PRB, 66, 024523 (2002); 73, 134502 (2006)] if the applied field B is much greater than the self-field B_s . At low fields $B < B_s$ the mean density of vortices across the strip becomes inhomogeneous, which obviously affects the field dependence of $I_c(B)$ even if the edge barrier is not taken into account. The self-field effects can thus contribute to the observed crossover of $I_c(H)$ below 50 mT which may not entirely result from the edge barrier pinning. The contribution of the self-field effects should be estimated and discussed.
2. On p.10, where it is said that " $I_c \propto 1/B$ is a fingerprint of edge pinning", a reference should be given. Moreover, the field dependence $I_c \propto 1/(B_0+B)$ is not exclusively characteristic of the edge pinning but has also been observed on different superconductors for more than 60 years (the so-called Kim model), see, e.g. a review by Campbell and Evetts or else.
3. Analysis of experimental data was done assuming uniform distributions of vortices, which is hardly the case. In the edge barrier pinning scenario the onset of resistivity correspond to vortices penetrating locally through materials defects at the edges and forming "rivers or trees" of magnetic flux. In this case the density of vortices is highly inhomogeneous and the flux density can be considerably lower than the assumed equilibrium value B/ϕ_0 , as the model calculations of this work also indicate. For instance, at low fields $B < B_s$, the density of vortices is mostly controlled by the dynamics of their penetration through edge defects rather than by the applied field B . The analysis of the experimental data based on the Bezuglyj-Shklovskij model likely underestimates the maximum velocities of vortices, which can also affect the values of superconducting parameters, particularly, the inelastic scattering times and the LO critical velocity extracted from the fits. These issues, particularly the role of edge defects, should be discussed.
4. Such important parameters of Nb-C as λ and ξ should be specified on pp 4 and 5, and the Pearl length λ^2/d be calculated explicitly and compared with the width of the film where it was first mentioned but not at the end of the paper around Table 1.
5. The description of the theoretical model in methods is not very informative and its relevance to the experimental part was not made clear. For instance, it was not explained why the TDGL eqs on p. 25 lack the inelastic scattering time constant which plays such an important role in the analysis of the experimental data based on the LO mechanism in the main text. Also the undefined expression $\text{div}(\mathbf{j}^{\text{Us}} - \mathbf{j}^{\text{GL}})$ is confusing, suggesting that the GL current somehow does not follow from the Usadel theory near T_c . Given a potentially diverse scientific readership of Nature Communication outside the photon detector community, it would be more appropriate to replace the cryptic $\text{div}(\mathbf{j}^{\text{Us}} - \mathbf{j}^{\text{GL}})$ with an explicit expression which readily follows from Eqs. 33 and 34 of Ref. 51.

Reviewer #2 (Remarks to the Author):

The work "Ultra-fast vortex motion in dirty Nb-C superconductor with a close-to-perfect edge barrier" by Dobrovolskiy et al. presents indirect experiment evidence of ultra-fast vortices in a Nb-C superconductor dominated by edge barrier. The theoretical analysis is based on a previous publication of one of the authors (Ref.[33]).

The paper is well written and structured, interesting, and scientifically sound. Although it certainly contains a good level of originality (sample fabrication, material, LLI in edge barrier dominated samples) and it deserves publication in some form, I consider that it does not really present sufficiently exciting results or will influence thinking in the field.

In addition, I believe the case presented in this manuscript would be reinforced if the results were compared with those obtained in a sample with roughness along its border so as to diminish the influence of the edge barriers. For instance introducing a small notch at the surface will reduce B_s .

The text is easy to read but here and there the English can be polished.

Response to the Referees' remarks

Referee #1

The Referee writes "This work reports extensive transport measurements of the flux flow instability which was interpreted in terms of the Larkin-Ovchinnikov (LO) instability affected by the edge barrier. This study extends many previous experimental investigations of the LO instability to a narrow film Nb-C film strips with controlled edge barrier properties which allowed the authors to extract the vortex velocities of the order of 10 km/s at the onset of instability. The results appear interesting and qualitative conclusions are supported by the experimental data, so this work in my opinion would be appropriate for Nature Communications."

We appreciate the Referee for the high evaluation of the quality of our studies.

"Yet the authors focused on a particular scenario, while disregarding other effects which can affect the valued of the parameters extracted from model fits, so the key assumptions should be clarified, as outlined below."

In the revision, we have carefully addressed all critical points raised by the Referee, as detailed next.

1. *"The field dependence $I_c(B) \propto B^{-a}$ with a $\approx 1/2$ observed at $B > 100$ mT has been often observed on superconducting films. It follows from many models of bulk pinning [see, e.g., PRB, 66, 024523 (2002); 73, 134502 (2006)] if the applied field B is much greater than the self-field B_s . At low fields $B < B_s$ the mean density of vortices across the strip becomes inhomogeneous, which obviously affects the field dependence of $I_c(B)$ even if the edge barrier is not taken into account. The self-field effects can thus contribute to the observed crossover of $I_c(H)$ below 50 mT which may not entirely result from the edge barrier pinning. The contribution of the self-field effects should be estimated and discussed."*

Just to prevent confusing, in our notation B_s is not the self-field but the field at which the surface barrier for vortex entry is suppressed at $I = 0$ and vortices enter the microstrip. With the estimate for the self-field $B_{\text{self}} = \mu_0 I / 2\pi w \ln(2w/d) \approx 10^{-2}$ mT at $I \sim 10^{-5}$ A we obtain $B_{\text{self}} \ll B_s \sim 10\text{-}20$ mT and, hence, the contribution of self-field effects to the observed crossover in $I_c(B)$ at $B \approx 10$ mT is negligibly small.

We agree with the Referee that there are many models of bulk pinning predicting $I_c(B) \sim B^{-1/2}$. For our consideration, however, a particular model of bulk pinning is not important as the main message of the fit to this law in the inset of Fig. 3 is that at $B > 100$ mT the bulk pinning also contributes to the critical current such that $I_c(B)$ can no longer be described by the edge-barrier pinning dependence $I_c(B) \sim B^{-1}$.

- In the revision, on page 11 we have amended the sentence "This dependence, **which follows from many models of bulk pinning**, can be explained by the increasing role of the intrinsic pinning at higher vortex densities at larger magnetic fields".

- In the revision, on page 8 we have added an estimate for the self field and mentioned that the observed crossover in $I_c(B)$ at about 10 mT can not result from the contribution of self-field effects, since the self field is much smaller than the fields at which the crossover occurs.

2. *"On p.10, where it is said that " $I_c \propto 1/B$ is a fingerprint of edge pinning", a reference should be given. Moreover, the field dependence $I_c \propto 1/(B_0+B)$ is not exclusively characteristic of the edge pinning but has also been observed on different superconductors for more than 60 years (the so-called Kim model), see, e.g. a review by Campbell and Evetts or else."*

We agree with the Referee that the $I_c \propto 1/B$ dependence alone can not be considered as a fingerprint of the edge pinning.

- In the revision, on page 11 we amended the sentence: " $I_c \propto 1/B$ **in conjunction with a linear decrease of $I_c(B)$ at low fields** is a fingerprint of the edge pinning" and added a reference to [Plourde et al. Influence of edge barriers on vortex dynamics in thin weak-pinning superconducting strips, Phys. Rev. B, **64**, 014503 (2001)].

3. *"Analysis of experimental data was done assuming uniform distributions of vortices, which is hardly the case. In the edge barrier pinning scenario the onset of resistivity correspond to vortices penetrating locally through materials defects at the edges and forming "rivers or trees" of magnetic flux. In this case the density of vortices is highly inhomogeneous and the flux density can be considerably lower than the assumed equilibrium value B/ϕ_0 , as the model calculations of this work also indicate. For instance, at low fields $B < B_s$, the density of vortices is mostly controlled by the dynamics of their penetration through edge defects rather than by the applied field B . The analysis of the experimental data based on the Bezuglyj-Shklovskij model likely underestimates the maximum velocities of vortices, which can also affect the values of superconducting parameters, particularly, the inelastic scattering times and the LO critical velocity extracted from the fits. These issues, particularly the role of edge defects, should be discussed."*

We agree with the Referee that at low fields the vortex distribution is not uniform, as is also illustrated in the insets in Fig. 5(a) and (b): There is a vortex dome and the number of vortices is smaller than the simple B/ϕ_0 estimate. If we use the correct (smaller) number of vortices, we deduce yet *higher* vortex velocities and, hence, yet *smaller* inelastic scattering times. Though accounting for the smaller number of vortices can somewhat improve the fits of v^* at low fields in Fig. 3(a), it cannot improve the estimates following from the Bezuglyj-Shklovskij model (which is justified for a uniform vortex distribution).

- In the revision, this statement has been added on page 18.

We would like to emphasize that our analysis in the manuscript is done within the framework of the edge-controlled instability model which **explicitly takes into account** the non-uniform distribution of vortices.

- In the revision, this statement has been added on page 16.

In our microstrips I_c is close to I_{dep} and this is why one can conclude that intrinsic defects do not influence I_c too much. At the same time, the absence of a smoothing of $I_c(B)$ at $B \ll B_{stop}$ suggests that a certain amount of edge defects can still be present in the microstrip. In general, this smoothing is expected for samples with ideal edge barriers due to the pair-breaking effect of the depairing current [Andratskii et al., Sov. Phys. JETP 38, 797 (1974)]. Accordingly, edge defects may influence not only I_c but also I^* in our microstrip. This effect was studied theoretically in Ref. [Vodolazov & Klapwijk, Phys. Rev. B 100, 064507 (2019)] where a photon-induced hot spot was playing the role of a defect. It was revealed that the defect practically does not influence I^* if the defect is far from the edge where vortices enter the strip. If the defect is located close to the edge where vortices enter the strip, a weak suppression of I^* is expected if the size of the defect is comparable with the width of the vortex-free region in the microstrip. Such a behavior is a direct consequence of the edge-controlled FFI, as discussed in detail in [Vodolazov & Klapwijk, Phys. Rev. B 100, 064507 (2019)].

- In the revision, this discussion has been added on page 11.

Please also see our reply to Referee 2 and the added discussion of a reference experiment for a microstrip with a notch at one of its edges on page 12 of the manuscript.

4. *"Such important parameters of Nb-C as λ and ξ should be specified on pp 4 and 5, and the Pearl length $\lambda^2/2d$ be calculated explicitly and compared with the width of the film where it was first mentioned but not at the end of the paper around Table 1."*

- In the revision, we have specified λ , ξ and the Pearl length at the place of first mentioning.

5. *"The description of the theoretical model in methods is not very informative and its relevance to the experimental part was not made clear. For instance, it was not explained why the TDGL eqs on p. 25 lack the inelastic scattering time constant which plays such an important role in the analysis of the experimental data based on the LO mechanism in the main text."*

• In the revision, on page 29 the following comment has been added: "The electron-photon scattering time enters the TDGL equation indirectly via the electron temperature T_e , whose dynamics is governed by $\tau_{e-ph} \sim \tau_0$ in the heat conductance equation. This is rather similar to the LO approach, where τ_{e-ph} enters the kinetic equation for the electron distribution function $f(E)$ (in our case this is the heat conductance equation for T_e) and $f(E)$ enters the GL equation in the LO work."

"Also the undefined expression $\text{div}(j^{Us} - j^{GL})$ is confusing, suggesting that the GL current somehow does not follow from the Usadel theory near T_c . Given a potentially diverse scientific readership of Nature Communication outside the photon detector community, it would be more appropriate to replace the cryptic $\text{div}(j^{Us} - j^{GL})$ with an explicit expression which readily follows from Eqs. 33 and 34 of Ref. 51."

• In the revision, on page 28 we have added the following comment: "At T not very close to T_c the Ginzburg-Landau equation for the superconducting current is not valid quantitatively and one needs to use the Usadel expression for j_s . If one uses it then one also should modify the TDGL equation because, for example, the ordinary TDGL equation leads to $\text{div} j^{GL} = 0$ in the stationary case, while one needs $\text{div} j^{Us} = 0$. Accordingly, by adding the term $\text{div}(j^{Us} - j^{GL})$ in the TDGL equation we provide $\text{div} j^{Us} = 0$. At $T \rightarrow T_c$ the modified TDGL equation reduces to the ordinary TDGL equation and $\text{div}(j^{Us} - j^{GL})$ goes to zero."

Reply to Referee #2

The Referee writes "The work "Ultra-fast vortex motion in dirty Nb-C superconductor with a close-to-perfect edge barrier" by Dobrovolskiy et al. presents indirect experiment evidence of ultra-fast vortices in a Nb-C superconductor dominated by edge barrier. The theoretical analysis is based on a previous publication of one of the authors (Ref.[33]). The paper is well written and structured, interesting, and scientifically sound."

We appreciate the Referee for the high evaluation of the quality of our studies.

"Although it certainly contains a good level of originality (sample fabrication, material, LLI in edge barrier dominated samples) and it deserves publication in some form, I consider that it does not really present sufficiently exciting results or will influence thinking in the field."

We think that this Referee's comment is a consequence of some unfortunate misunderstanding. The exciting physics emerging from fast vortex motion has been explicitly emphasized in the introduction: The physics of moving vortex matter is getting especially rich when the vortex velocity exceeds the velocity (3-5 km/s) of other possible excitations in the system. For instance, such high velocities are required for the excitation of short-wavelength spin-waves by the Cherenkov mechanism in the rapidly developing domains of fluxon magnonics and magnon spintronics.

Che et al. *Nat. Commun.* 11, 1445 (2020), <https://www.nature.com/articles/s41467-020-15265-1>

Dobrovolskiy et al., *Nat. Phys.* 15, 477 (2019), <https://www.nature.com/articles/s41567-019-0428-5>

Yu et al., *Nat. Commun.* 7, 11255 (2016), <https://www.nature.com/articles/ncomms11255>

Chumak et al., *Nat. Phys.* 11, 453 (2015), <https://www.nature.com/articles/nphys3347>

However, the phenomenon of flux-flow instability prevents the exploration of this intriguing physics and sets practical limits for the use of vortices in various applications. To suppress instability, a superconductor should exhibit a rarely achieved combination of properties: weak volume pinning, close-to-depairing critical current, and fast heat removal to the substrate. In our work, we successfully meet this challenge and **demonstrate vortex velocities 10-15 km/s in a uniform system for the first time.**

Several important aspects are addressed in our manuscript at the same time, namely

Fundamental physics - new type of flux-flow instability allows for up to 15 km/s vortex velocities. The instability mechanism is different from the widely used theories. We also discuss their applicability to the studied system;

Materials science - direct-write nanofabrication is used to create close-to-perfect edge barriers. And it is the quality of the edge which in conjunction with a fast rate of the relaxation of disequilibrium allows for ultra-fast vortex dynamics;

Applications - Nb-C exhibits a very short relaxation time and a close-to-depairing critical current, representing a new directly written material for fast single-photon detectors which can be readily on-chip and on-fiber integrated taking advantage of the mask-less nanofabrication;

We are highly confident that our results will be interesting to a broad readership of Nature Communications and will influence thinking in the field.

"In addition, I believe the case presented in this manuscript would be reinforced if the results were compared with those obtained in a sample with roughness along its border so as to diminish the influence of the edge barriers. For instance introducing a small notch at the surface will reduce B_s ."

• We have addressed this Referee's suggestion in the revision. Figure 4 with the $I_c(B)$ and $I^*(B)$ dependences for the microstrip with a notch has been added as a direct proof of the edge-controlled mechanism of the instability in our work. The description of this experiment has been added on pages 12-13.

"The text is easy to read but here and there the English can be polished."

• We have carefully proofread the manuscript and polished a series of phrases in the revision.

REVIEWER COMMENTS

Reviewer #1 (Remarks to the Author):

The revised version has been improved and some of the points of my previous report have been addressed. In my opinion this work is interesting and addresses important issues of superfast dynamics of vortices so I could recommend it for publication in Nat. Comm. Although the arguments of the authors appear convincing, I think that they should not ascribe their results exclusively to the edge-barrier mechanism but mention other possibilities. As I pointed out in my first report, the well-known $I_c \propto 1/(B_0+B)$ of the Kim model of bulk pinning fits the observations of this work as well, giving a linear dependence of $I_c(B)$ on $B < B_0$ and $I_c \propto 1/B$ at $B \gg B_0$. (Kim and Stephen in Superconductivity v. 2, ed. by Parks, p. 1107 (1969); Campbell and Evets, Adv. Phys. 21, 199 (1972)]. Perhaps, the authors should soften their categorical claim that their observations are "fingerprints of the edge barrier model...", replacing it with something like "the totality of our experimental data indicates ..." while mentioning other potential scenarios along with corresponding references. For instance, it would be worth mentioning in the introduction and discussion sections that jumps on the V-I curves could also result from the hotspot formation unrelated to the LO instability. The Joule heating in these films at low T can be pretty strong: at 4.2K, $I = 20 \mu\text{A}$ and $V = 5 \text{ mV}$, the heat flux from the film $q = IV/lw = 15 \text{ kW/m}^2$. For typical values of the Kapitza interface thermal conductance of the order of 10 kW/m^2 and not very effective diffusive heat removal across the Si substrate with thermal conductivity some 2 orders of magnitude smaller than those of good metals, the Joule heating could cause phonon overheating of a few K.

The description of the TDGL equations in the Methods is still incomplete as $\text{div}(J^U_s - J^G_L)$ was not specified. The whole point of giving these equations in the Methods (particularly in this journal aimed at a more general physics audience) is that the reader can see the full set of equations used in the simulations without going to original papers. Using the rather technical notions of J^U_s and J^G_L without explaining what they are and directing the reader to Ref. 32 defies the purpose of giving these eqs in the first place. The authors could just have shown the results in Fig. 5 and stated that they were obtained using the model eqs of Ref. 32.

Response to the Referees' remarks

Referee #1

The Referee writes "The revised version has been improved and some of the points of my previous report have been addressed. In my opinion this work is interesting and addresses important issues of superfast dynamics of vortices so I could recommend it for publication in Nat. Comm."

We appreciate the Referee for the high evaluation of the revision.

"Although the arguments of the authors appear convincing, I think that they should not ascribe their results exclusively to the edge-barrier mechanism but mention other possibilities. As I pointed out in my first report, the well-known $I_c \propto 1/(B_0+B)$ of the Kim model of bulk pinning fits the observations of this work as well, giving a linear dependence of $I_c(B)$ on $B < B_0$ and $I_c \propto 1/B$ at $B \gg B_0$. (Kim and Stephen in Superconductivity v. 2, ed. by Parks, p. 1107 (1969); Campbell and Evets, Adv. Phys. 21, 199 (1972)). Perhaps, the authors should soften their categorical claim that their observations are "fingerprints of the edge barrier model...", replacing it with something like "the totality of our experimental data indicates ..." while mentioning other potential scenarios along with corresponding references."

We would like to note that the parameter B_0 in the dependence $I_c \propto 1/(B_0+B)$ in the bulk pinning model is a phenomenological one. By contrast, in our case the field B_{stop} is a well-defined field (via parameters of the microstrip) and it perfectly fits the experimental data. We note that in our samples $I_c(0) \sim I_{\text{dep}}$ and such a high $I_c(0)$ cannot be explained by any existing theory of bulk pinning (usually j_c is at least 10 times smaller than j_{dep}). Furthermore, the dependence $I_c(B) \propto 1/(B_0+B)$ can be viewed as a quasi-linear dependence at $B < 0.1B_0$ only and $I_c \sim 1/B$ at $B \geq 3B_0$ cannot fit our $I_c(B)$ at any B_0 . This makes us believe that our $I_c(B)$ is well fitted by the theory for edge barrier for vortex entry. Two additional checks (measurements for a microstrip with an edge defect in Fig. 4a and the I - V curve for a narrower microstrip in Fig. 6a) further corroborate the edge-barrier mechanism.

Change made: We have added a reference to Kim and Stephen and replaced the phrase "fingerprints of the edge barrier model..." with "Though the dependence $I_c \propto 1/(B_0 + B)$ of the Kim model of bulk pinning [37] could, in principle, describe a linear dependence $I_c(B)$ at fields below some field B_0 and $I_c \propto 1/B$ at $B \gg B_0$, the totality of our experimental data, to be discussed in what follows, indicates the dominating role of the edge mechanism of vortex pinning in the studied sample at $B < 100$ mT". A reference to Kim and Stephen in Superconductivity v. 2, ed. by Parks, p. 1107 (1969) has been added.

"For instance, it would be worth mentioning in the introduction and discussion sections that jumps on the V - I curves could also result from the hotspot formation unrelated to the LO instability. The Joule heating in these films at low T can be pretty strong: at 4.2K, $I = 20$ μ A and $V = 5$ mV, the heat flux from the film $q = IV/lw = 15$ kW/m². For typical values of the Kapitza interface thermal conductance of the order of 10 kW/m² and not very effective diffusive heat removal across the Si substrate with thermal conductivity some 2 orders of magnitude smaller than those of good metals, the Joule heating could cause phonon overheating of a few K."

We would like to note that in the hot spot model, the product I^*V^* does not depend on B whereas in our experiment it does. Therefore, our experimental data cannot be explained within the framework of the hot spot model. At the same time, in our model, both Joule heating and the LO-like mechanism are taken into account.

Change made: Just before the Discussion section we have added the phrase "We would like to note that jumps on the I - V curves could also result from the hot spot formation unrelated to the FFI. However, in the hot spot model the dissipated power at the instability point I^*V^* does not depend on B whereas in our experiment it does, as is peculiar to the FFI [3,24,25]. Accordingly, the obtained experimental data cannot be explained by the hot spot model alone, and both Joule heating and the LO-like FFI mechanism are taken into account in the models discussed next".

"The description of the TDGL equations in the Methods is still incomplete as $\text{div}(\mathbf{j}^{\text{Us}} - \mathbf{j}^{\text{GL}})$ was not specified. The whole point of giving these equations in the Methods (particularly in this journal aimed at a more general physics audience) is that the reader can see the full set of equations used in the simulations without going to original papers. Using the rather technical notions of \mathbf{j}^{Us} and \mathbf{j}^{GL} without explaining what they are and directing the reader to Ref. 32 defies the purpose of giving these eqs in the first place. The authors could just have shown the results in Fig. 5 and stated that they were obtained using the model eqs of Ref. 32."

We thank the referee for this remark.

Change made: In the revision, we have specified \mathbf{j}^{Us} and \mathbf{j}^{GL} in the Methods so that the reader can see the full set of equations used in the simulations without going to original papers.